# NFAT activation by FKBP52 promotes cancer cell proliferation by suppressing p53

Shunsuke Hanaki[1], Makoto Habara[1], Haruki Tomiyasu[1], Yuki Sato[1], Yosei Miki[1], Takahiro Masaki[1], Shusaku Shibutani[2], Midori Shimada[1,3]

**FK506-binding protein 52 (FKBP52) is a member of the FKBP family of proline isomerases. FKBP52 is up-regulated in various cancers and functions as a positive regulator of steroid hormone receptors. Depletion of FKBP52 is known to inhibit cell proliferation; however, the detailed mechanism remains poorly understood. In this study, we found that FKBP52 depletion decreased *MDM2* transcription, leading to stabilization of p53, and suppressed cell proliferation. We identified NFATc1 and NFATc3 as transcription factors that regulate *MDM2*. We also found that FKBP52 associated with NFATc3 and facilitated its nuclear translocation. In addition, calcineurin, a well-known $Ca^{2+}$ phosphatase essential for activation of NFAT, plays a role in *MDM2* transcription. Supporting this notion, *MDM2* expression was found to be regulated by intracellular $Ca^{2+}$. Taken together, these findings reveal a new role of FKBP52 in promoting cell proliferation via the NFAT-MDM2-p53 axis, and indicate that inhibition of FKBP52 could be a new therapeutic tool to activate p53 and inhibit cell proliferation.**

## Introduction

FKBP52 is an immunophilin, is a member of the FKBP family, and is encoded by FKBP4. FKBP52 has been reported to be associated with cancer, psychiatric disorders, hormone-dependent diseases, and stress-related pathologies (Zgajnar et al, 2019). The two main functions of FKBP52 are as follows. First, it promotes the nuclear translocation of target proteins, such as steroid hormone receptors, by binding to dynein (Silverstein et al, 1999; Galigniana et al, 2001, 2010; Mackenzie et al, 2006; Mikenberg et al, 2007; Tatro et al, 2009; Erlejman et al, 2014; Jeong et al, 2016). Second, FKBP52 stabilizes target proteins by interacting with HSP90 by functioning as a co-chaperone (Jeong et al, 2016; Habara et al, 2022). Previous studies have shown that FKBP52 contributes to cancer cell proliferation and that FKBP52 expression is up-regulated in several cancers. These

results indicate that high FKBP52 expression is linked with poor prognosis (Ward et al, 1999; Lin et al, 2007; Liu et al, 2010; Mange et al, 2019; Meng et al, 2020; Liu & Gao, 2021; Zong et al, 2021; Habara et al, 2022; Maeda et al, 2022; Zhu et al, 2022). The mechanisms by which FKBP52 promotes cell proliferation in cancer cells include stabilization of ERα (Habara et al, 2022), dimerization of ARs (Maeda et al, 2022), and promotion of nuclear translocation of RelA (Erlejman et al, 2014) and IKK complex formation (Zong et al, 2021). FKBP52 has also been reported to activate the PI3K/AKT pathway, via unknown mechanisms (Mange et al, 2019; Meng et al, 2020). Taken together, these reports suggest that FKBP52 positively regulates cancer cell proliferation. However, the mechanism by which FKBP52 regulates proliferation remains unclear.

The NFAT transcription factors consist of five members, that is, NFATc1-c4 and NFAT5. These factors, except for NFAT5, are activated by intracellular $Ca^{2+}$ and play a crucial role in promoting the transcription of genes related to immune response (Lopez-Rodriguez et al, 1999; Fric et al, 2012). An increase of intracellular $Ca^{2+}$ levels triggers the activation of a downstream phosphatase known as calcineurin (Masaki & Shimada, 2022; Masaki et al, 2023a, 2023b). Calcineurin, in turn, facilitates dephosphorylation of NFAT and promotes nuclear translocation of NFAT. Beyond their roles in immune function, NFAT has been implicated in various cancer types, with elevated expression levels observed in a range of malignancies (Buchholz et al, 2006; Zhang et al, 2007, 2012; Chen et al, 2011; Le Roy et al, 2012; Pei et al, 2012; Tie et al, 2013).

p53 is a major tumor suppressor protein encoded by *TP53*. p53 is activated by various stress signals, such as DNA damage or excessive replication stimulus, and promotes the transcription of downstream target genes (Hernandez Borrero & El-Deiry, 2021). p53 target genes play a role in inducing cell cycle arrest, DNA damage repair, and apoptosis. p53 activity is inhibited by the ubiquitin ligase MDM2, which binds to the DNA-binding domain of the p53, thereby inhibiting the binding of p53 to DNA (Chen et al, 1993; Oliner et al, 1993; Picksley et al, 1994), and promotes degradation by ubiquitinating the p53 protein (Honda et al, 1997). In this regard, high MDM2 expression promotes p53 degradation and enhances cancer cell proliferation.

---

[1]Department of Veterinary Biochemistry, Yamaguchi University, Yamaguchi, Japan   [2]Department of Veterinary Hygiene, Yamaguchi University, Yamaguchi, Japan   [3]Department of Molecular Biology, Nagoya University, Graduate School of Medicine, Nagoya, Japan

Correspondence: shimada@med.nagoya-u.ac.jp

In this study, we analyzed comprehensive transcriptome data from FKBP52-depleted cells to elucidate the mechanism by which FKBP52 promotes cell proliferation. FKBP52 promotes the nuclear translocation of NFAT, which in turn accelerates *MDM2* transcription and p53 degradation.

# Result

### p53 is up-regulated in FKBP52-KD cells

We found that depletion of FKBP52 in MCF7 breast cancer cells expressing WT p53 markedly inhibited proliferation (Habara et al, 2022). To elucidate this mechanism, we performed RNA-seq analysis and GSEA in FKBP52-depleted cells. The results showed that the expression of p53 signaling pathway–related genes was significantly up-regulated in FKBP52-depleted cells (Fig 1A and B and Table S1). These results suggest that FKBP52 depletion induces p53 activation. Therefore, we examined the protein expression of p53 and its major target, p21, in FKBP52-depleted MCF7 cells. The results showed that the expression of p53 and p21 significantly increased in FKBP52-depleted cells (Fig 1C). In addition, the protein expression of PUMA and GADD45A, which are targets of p53, also increased in FKBP52-depleted cells (Fig S1A). Depletion of FKBP52 did not increase γH2AX levels, indicating that p53 activation is not induced by DNA damage. We also confirmed FKBP52 depletion increased the abundance of p53 and p21 in HCT116 cells (Fig S1B). To examine the transcriptional activity of p53, we performed a luciferase reporter assay. We found that p53 transcriptional activity was significantly increased in FKBP52-depleted cells (Fig 1D). In addition, *TP53* mRNA expression was analyzed. There was no significant increase in *TP53* mRNA expression in FKBP52-depleted cells (Fig 1E), suggesting that the expression of p53 is regulated at the protein level. In HCT116 cells, FKBP52 depletion did not affect *TP53* mRNA expression (Fig S1C). These results indicate that p53 is activated by FKBP52 depletion.

### FKBP52 depletion induces p53-dependent cell growth arrest and apoptosis

Because p53 is activated in FKBP52-deficient cells, we examined whether the decrease in proliferation observed in FKBP52-deficient cells was dependent on p53. The results showed that cells doubly deficient in FKBP52 and p53 recovered the decrease in proliferation caused by FKBP52 deficiency and exhibited a growth rate comparable to that of cells deficient in p53 (Fig 2A).

Next, the *TP53* gene region was deleted in HCT116 cells using CRISPR/Cas9 and several cells showed no expression of p53 protein (Fig S2A). Among them, p53-KO (clone 10) harbored CRISPR/Cas9-induced deletion and insertion of the puromycin resistance gene (Fig S2B); therefore, clone 10 was used for further analysis as HCT116-p53-KO. As expected, depletion of FKBP52 inhibited proliferation in a p53-dependent manner (Fig S2C). Similar results were confirmed in HCT116 cells depleted of FKBP52 and p53 using shRNA (Fig S2D). To clarify the effect of FKBP52 depletion on the cell cycle, we performed a cell cycle analysis using flow cytometry. The results showed that the percentage of sub-G1 cells, which indicates dead cells, was significantly increased in FKBP52-depleted cells (Figs 2B

and S2E). Sub-G1 cells were decreased in FKBP52 and p53 double-depleted cells compared with that in cells depleted of FKBP52 alone. The percentages of S- and G2/M-phase cells were also significantly decreased in FKBP52-depleted cells. To test whether FKBP52 depletion affects the mRNA expression of p53 target genes, we measured the expression of *CDKN1A* (encoding p21), *GADD45A*, and *BBC3* (encoding PUMA). The mRNA expression of these three genes significantly increased in FKBP52-depleted cells (Fig 2C). Similar experiments using HCT116 cells showed a significant increase in p53 target genes in FKBP52-depleted cells (Fig S2F and G). In addition, double depletion of FKBP52 and p53 did not increase the mRNA expression of these three genes (Fig 2C). These results indicate that depletion of FKBP52 suppresses cell proliferation and causes cell death in a p53-dependent manner.

### p53 is stabilized in FKBP52-KD cells

To determine the cause of the increase in p53 protein expression upon FKBP52 depletion, we examined *TP53* mRNA expression but did not find any decrease in the *TP53* mRNA level (Fig 1E). We therefore focused on p53 protein stability. An assay using cycloheximide revealed that the half-life of p53 was extended in FKBP52-depleted cells (Figs 3A and S3A). This indicates that the depletion of FKBP52 stabilizes the p53 protein.

Therefore, we focused on MDM2, which is a major ubiquitin ligase regulating p53 protein degradation. MDM2 protein expression significantly decreased in FKBP52-depleted cells (Fig 3B). A similar phenomenon was observed in the HCT116 cells (Fig S1B). To explore the reason for this decrease in MDM2 protein expression, we measured the mRNA of *MDM2*. The results showed that *MDM2* mRNA was significantly decreased in FKBP52-depleted cells (Fig 3C). A similar experiment using HCT116 cells revealed a significant decrease in *MDM2* mRNA expression (Fig S3B). To comprehensively examine whether FKBP52 affects *MDM2* mRNA expression, we determined the correlation coefficient between FKBP52 protein and *MDM2* mRNA expression using the breast cancer proteome database (Krug et al, 2020). This analysis revealed a positive correlation between the FKBP52 protein level and *MDM2* mRNA expression (Fig 3D). These results indicate that in FKBP52-depleted cells, an *MDM2* mRNA decrease reciprocally increases the p53 protein stability.

### NFAT promotes *MDM2* transcription

We searched for candidate transcription factors for *MDM2* using the following criteria: (1) a factor that binds to the promoter region of *MDM2* using the GeneCards database, (2) a factor that decreases *MDM2* mRNA upon knockdown of a transcription factor using the KnockTF database, and (3) a factor that has a positive correlation of 0.2 or more with *MDM2* mRNA using TCGA database. We identified NFATc3 as a candidate transcription factor that satisfies all three conditions (Fig 4A). NFATc4 and NFAT5 were excluded because of their low expression in MCF7 cells (Uhlen et al, 2015) (https://www.proteinatlas.org/ENSG00000100968-NFATC4/cell+line) (https://www.proteinatlas.org/ENSG00000102908-NFAT5/cell+line); therefore, we investigated the importance of NFATc1, NFATc2, and NFATc3 for regulation of the expression level of MDM2. In MCF7 cells, depletion of NFATc1 and NFATc3 significantly

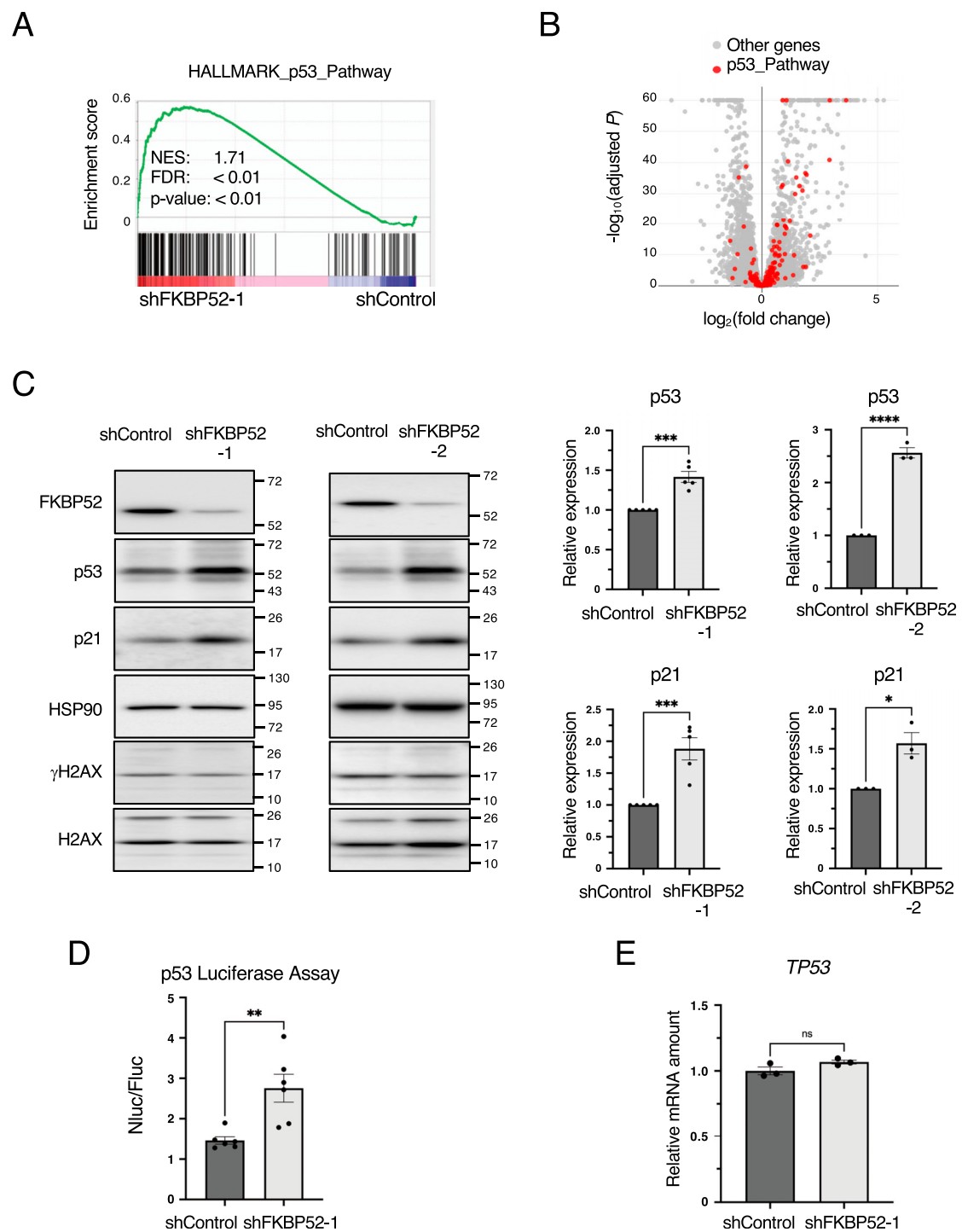

**Figure 1. Depletion of FKBP52 activates p53.**

**(A)** Enrichment plot of HALLMARK_p53_Pathway in the FKBP52-depleted MCF7 cells compared with the control was generated using GSEA software. NES, normalized enrichment score; FDR, false discovery rate. Three biological replicates were analyzed. **(B)** Volcano plot comparing the gene expression levels in shControl MCF7 cells and shFKBP52. Genes included in the HALLMARK_p53_Pathway are indicated by red dots, and other genes are indicated by gray dots. The y-axis denotes –log10 adjusted $P$-values, whereas the x-axis shows $\log_2$ fold-change values. **(C)** MCF7 cells were cultured in the presence of Dox for 3 d to knock down FKBP52 or luciferase (shControl) using tetracycline-inducible shRNA. The cells were collected, and the total cell extracts were subjected to immunoblotting using the indicated antibodies. Signals were quantified using Image Lab software. The relative expression was calculated by dividing the band intensities of p53 and p21 by those of HSP90. The bar chart shows the mean ± SEM of three independent experiments. Each value was tested using a $t$ test. *$P < 0.05$, **$P < 0.01$, ****$P < 0.0001$. **(D)** HEK293T cells were transfected with pNL containing the p53 response sequence and pGL4.53 containing the PGK promoter, in the presence of Dox for 3 d. NanoLuc luciferase activity was normalized to the FLuc activity. The bar chart shows the mean ± SEM of six independent experiments. Values were tested using a $t$ test. **$P < 0.01$. **(E)** RT–qPCR analysis of *TP53* expression in the control and FKBP52-depleted MCF7 cells. The bar chart shows the mean ± SEM of three independent experiments. Each value was tested using a $t$ test.

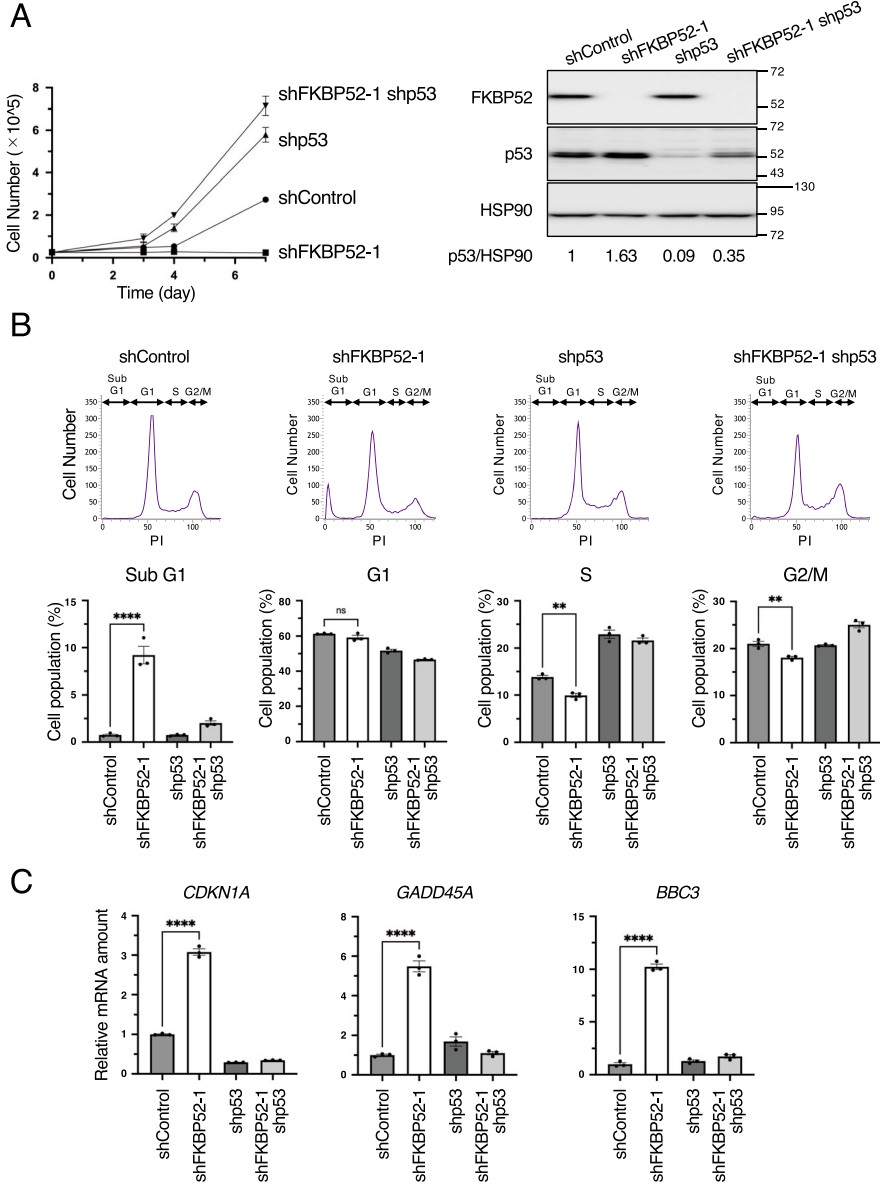

**Figure 2. FKBP52 depletion suppresses cell proliferation and promotes apoptosis in a p53-dependent manner.**
**(A)** MCF7 cells were cultured in the presence of Dox for indicated days to knock down FKBP52, p53, or luciferase (shControl) using tetracycline-inducible shRNA. Cells were cultured and collected, and cell numbers were counted. Data are presented as the mean ± SEM of three independent experiments (left). MCF7 cells were cultured in the presence of Dox for 3 d to knock down FKBP52, p53, or luciferase (shControl) using tetracycline-inducible shRNA. The cells were collected, and the total cell extracts were subjected to immunoblotting using the indicated antibodies. Signals were quantified using Image Lab software. The relative expression was calculated by dividing the band intensity of p53 by that of HSP90 (right). **(B)** Cell cycle distribution of control and FKBP52-depleted MCF7 cells was verified using FACS analysis. The cells were stained with PI. The proportion of cells in each phase of the cell cycle was quantified. The bar chart shows the mean ± SEM of three independent experiments. Values were tested using a $t$ test. **$P < 0.01$, ****$P < 0.0001$. **(C)** RT–qPCR analysis of $p21$, $GADD45A$, and $PUMA$ expression in MCF7 cells. The bar chart shows the mean ± SEM of three independent experiments. Each value was tested using a $t$ test.

decreased both MDM2 protein and mRNA expressions (Figs 4B and C and S4A and B). In contrast, NFATc2 depletion did not decrease MDM2 protein expression. As NFAT is positively regulated by calcineurin, calcineurin depletion is expected to inactivate NFAT and in turn decrease MDM2 protein and mRNA levels. Consistent with this notion, depletion of calcineurin in MCF7 cells resulted in a significant decrease in MDM2 protein and mRNA (Figs 4D and E and S4C). As the calcineurin/NFAT pathway was found to be important for $MDM2$ mRNA transcription, we used INCA-6, a chemical that inhibits calcineurin/NFAT binding and suppresses NFAT. INCA-6 treatment significantly decreased $MDM2$ mRNA expression (Fig 4F). Treatment with A23187, a $Ca^{2+}$ ionophore that activates calcineurin, significantly increased $MDM2$ mRNA expression in a concentration-dependent manner (Fig 4G). These results indicate that $MDM2$ mRNA is transcribed via the calcineurin/NFAT pathway.

## FKBP52 is involved in the translocation and transcriptional activity of NFAT

To investigate whether FKBP52 is involved in $MDM2$ transcription induced by NFAT, a luciferase reporter assay for NFAT was performed using FKBP52-depleted cells. The results showed that in control cells, A23187 treatment increased the transcriptional activity of NFAT by approximately fivefold, whereas in cells depleted of FKBP52, the increase was only twofold (Fig 5A). This indicates that FKBP52 is important for the transcriptional activity of NFAT. As FKBP52 is involved in the nuclear translocation of transcription factors, we investigated its effect on the nuclear translocation of NFAT. The results showed that in FKBP52-depleted cells, cytoplasmic NFATc1 and NFATc3 expression was increased, whereas those in the nucleus and chromatin were

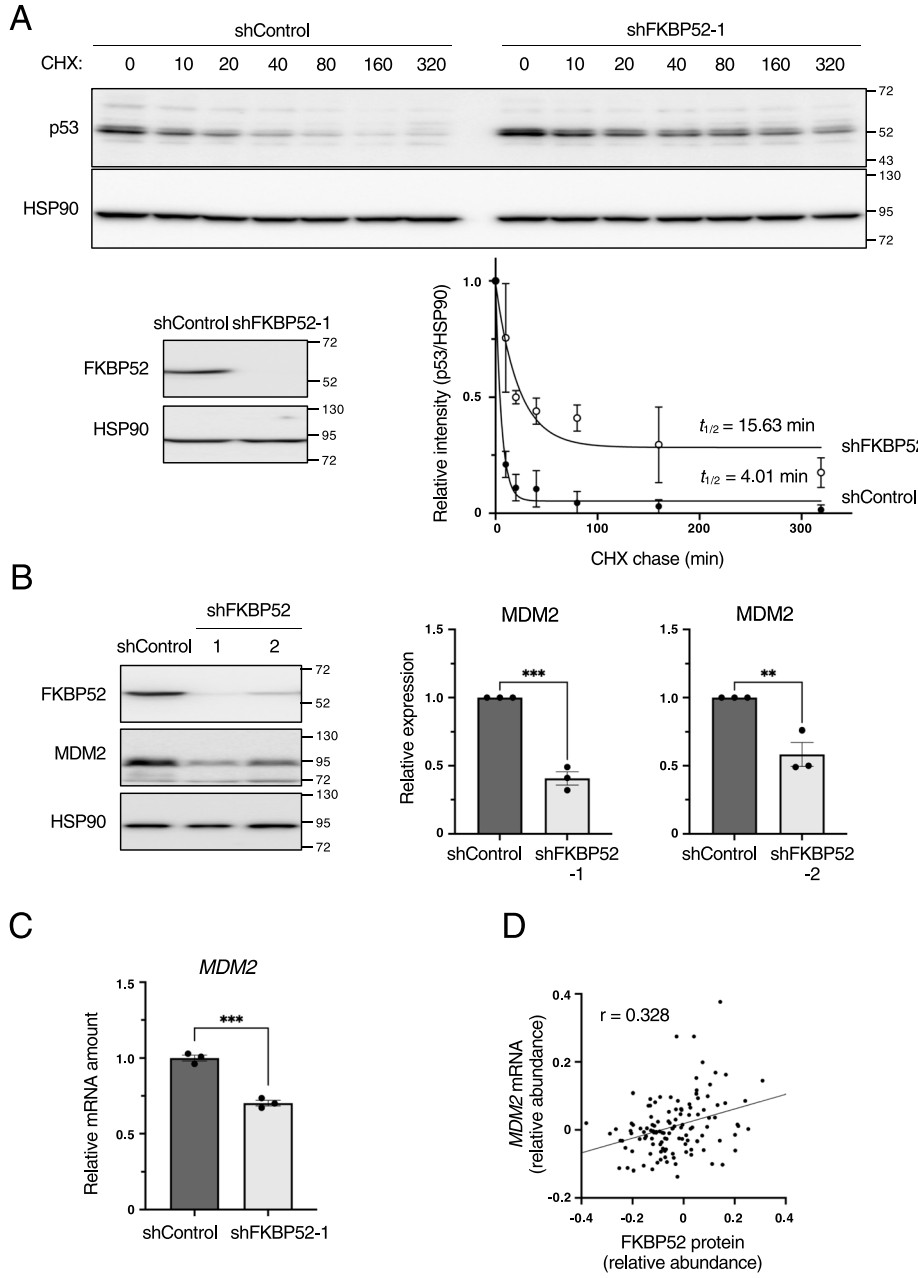

**Figure 3.  FKBP52 depletion suppresses MDM2 expression and stabilizes p53.**
**(A)** MCF7 cells expressing shRNAs were cultured in the presence of Dox for 2 d and then treated with 50 µg/ml cycloheximide for the indicated time intervals. Total cell lysates were subjected to immunoblotting using the indicated antibodies. The results are expressed as the mean ± SEM of two independent experiments. Half-life was calculated using GraphPad Prism version 9 (GraphPad Software). **(B)** MCF7 cells were cultured in the presence of Dox for 3 d to knock down FKBP52 or luciferase (shControl) using tetracycline-inducible shRNA. The cells were collected, and the total cell extracts were subjected to immunoblotting using the indicated antibodies. Signals were quantified using Image Lab software. The relative expression was calculated by dividing the band intensity of MDM2 by that of HSP90. The bar chart shows the mean ± SEM of three independent experiments. Each value was tested using a $t$ test. \*\*$P < 0.01$, \*\*\*$P < 0.001$. **(C)** RT–qPCR analysis of *MDM2* expression in the control and FKBP52-depleted MCF7 cells. The bar chart shows the mean ± SEM of three independent experiments. Each value was tested using a $t$ test. **(D)** Scatter plot of FKBP52 protein and *MDM2* mRNA expression. Dots indicate a single sample. r indicates the correlation coefficient, and the straight line indicates the regression lines.

decreased (Figs 5B and S5A). A similar phenomenon was observed in HCT116 cells (Fig S5B). We also found that FKBP52 is present in the nuclear and chromatin fraction but in relatively small amounts (Fig 5B). In addition, protein levels of p53 were increased in all fractions, suggesting that FKBP52 did not affect p53 localization. Notably, FKBP52 and NFATc3 were co-localized by immunostaining (Fig 5C). The effect of FKBP52 depletion on $Ca^{2+}$ concentration was investigated using the $Ca^{2+}$ indicator Fura-2AM. The result revealed that FKBP52 depletion does not affect the steady-state $Ca^{2+}$ concentration (Fig S5C). These results indicate that FKBP52 depletion inhibits the nuclear translocation of NFATc1 and NFATc3 without affecting intracellular $Ca^{2+}$ levels.

Chromatin immunoprecipitation was performed to determine the amount of NFATc1 and NFATc3 on the *MDM2* promoter region. The binding of NFATc1 and NFATc3 to the *MDM2* promoter region was decreased in FKBP52-depleted cells (Fig 5D). The binding of FKBP52 to the *MDM2* promoter region was also decreased. The results indicate that not only NFAT but also FKBP52 is located in the promoter region of *MDM2*. As FKBP52 and NFAT were shown to be located in the promoter region of *MDM2*, the binding of FKBP52 and NFATc3 was also examined. The results showed that FKBP52 and NFATc3 are associated with each other (Fig 5E). These results indicate that FKBP52 enhances transcriptional activity and promotes *MDM2* transcription by promoting nuclear translocation of NFATc1 and NFATc3.

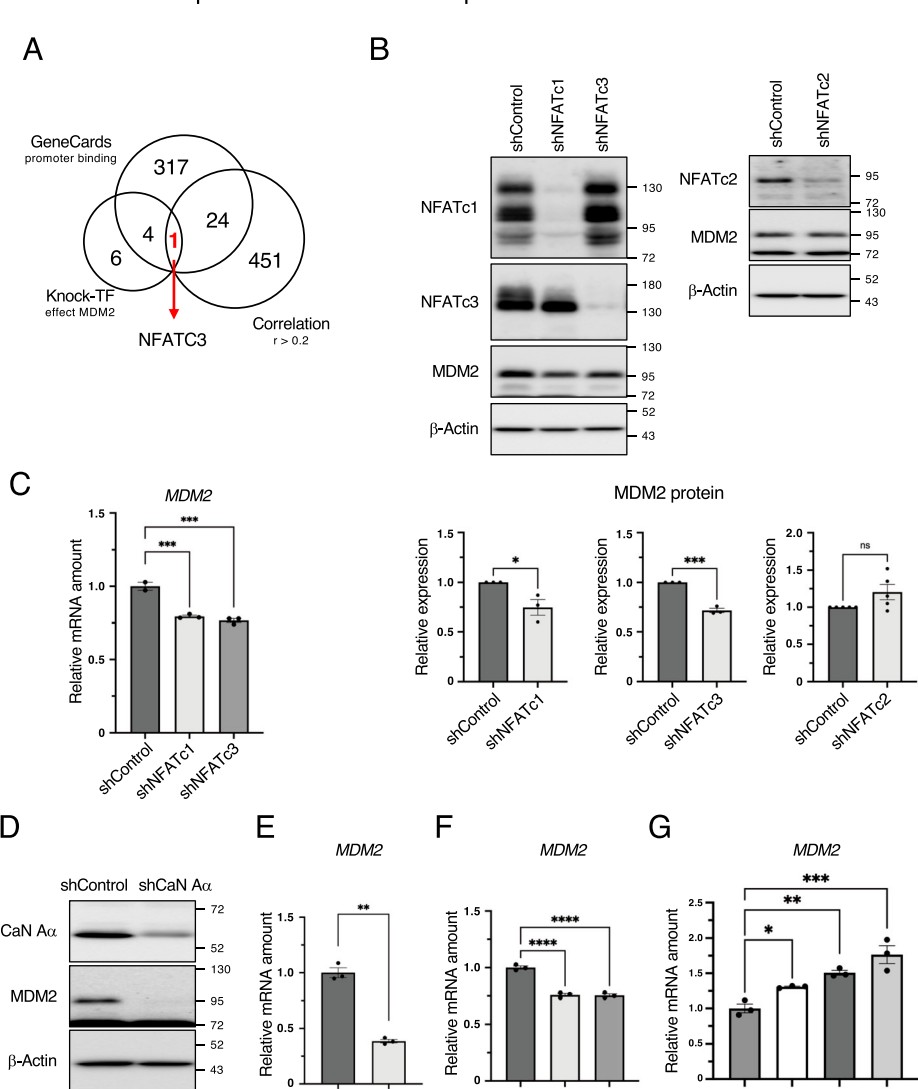

**Figure 4. Calcineurin/NFAT pathway promotes MDM2 transcription.**
**(A)** Venn diagram showing the number of molecules that bind to the promoter region of MDM2 in GeneCards, whose knockdown decreases *MDM2* mRNA in KnockTF, and have a correlation coefficient greater than 0.2 with *MDM2* expression in TCGA. **(B)** MCF7 cells were cultured in the presence of Dox for 3 d to knock down NFATc1, NFATc2, NFATc3, or luciferase (shControl) using tetracycline-inducible shRNA. The cells were collected, and the total cell extracts were subjected to immunoblotting using the indicated antibodies. Signals were quantified using Image Lab software. The relative expression was calculated by dividing the band intensity of MDM2 by that of $\beta$-actin. The bar chart shows the mean ± SEM of three independent experiments. Each value was tested using a *t* test. *$P$ < 0.05, ***$P$ < 0.001. **(C)** RT–qPCR analysis of *MDM2* expression in control and NFATc1- or NFATc3-depleted MCF7 cells. The bar chart shows the mean ± SEM of three independent experiments. Each value was tested using one-way ANOVA, followed by Dunnett's test. **(D)** MCF7 cells were cultured in the presence of Dox for 3 d to knock down CaN A$\alpha$ or luciferase (shControl) using tetracycline-inducible shRNA. The cells were collected, and the total cell extracts were subjected to immunoblotting using the indicated antibodies. **(E)** RT–qPCR analysis of *MDM2* expression in the control and CaN-A$\alpha$–depleted MCF7 cells. The bar chart shows the mean ± SEM of three independent experiments. Each value was tested using a *t* test. **(F)** RT–qPCR analysis of *MDM2* expression in the control and INCA-6–treated MCF7 cells. INCA-6 was treated at the indicated concentrations for 27 h. The bar chart shows the mean ± SEM of three independent experiments. Each value was tested using one-way ANOVA, followed by Dunnett's test. **(G)** RT–qPCR analysis of *MDM2* expression in the control and A23187-treated MCF7 cells. A23187 was added at the indicated concentration for 8 h. The bar chart shows the mean ± SEM of three independent experiments. Each value was tested using one-way ANOVA, followed by Dunnett's test.

## Discussion

In this study, we revealed a molecular mechanism by which FKBP52 promoted cell proliferation. We identified NFATc1 and NFATc3 as transcription factors regulating *MDM2*, and found that FKBP52 promoted the nuclear translocation of NFATc1 and NFATc3, which in turn promoted *MDM2* transcription and repressed p53 (Fig 5F). Previous studies have shown that FKBP52 is up-regulated in prostate (Lin et al, 2007; Periyasamy et al, 2007), breast (Ward et al, 1999; Mange et al, 2019; Habara et al, 2022), colon (Liu & Gao, 2021), lung (Zong et al, 2021), liver (Liu et al, 2010), ovarian (Habara et al, 2022), and leukemia (Habara et al, 2022) cancers, suggesting that FKBP52 may be an effective target for cancer therapy. Detailed mechanisms have been identified in prostate, breast, and lung cancer, such as AR dimerization (Maeda et al, 2022), ER$\alpha$ stabilization (Habara et al, 2022), and NF-$\kappa$B activation (Erlejman et al,

2014; Zong et al, 2021), respectively. In addition, the activation of the PI3K/AKT pathway has been reported, revealing a role of FKBP52 in cancer cell proliferation (Mange et al, 2019; Meng et al, 2020). However, studies showing an association with the cancer suppressor gene p53 are limited. To our knowledge, this is the first report to show the effects of FKBP52 and p53 on cell proliferation.

FKBP52 is known to promote the nuclear translocation of transcription factors, such as glucocorticoid receptor (Tatro et al, 2009), mineralocorticoid receptor (Galigniana et al, 2010), and RelA (Erlejman et al, 2014), and hTERT (Jeong et al, 2016) by binding to dynein. p53 is also known to bind to dynein for nuclear translocation (Giannakakou et al, 2000), and it has been reported that FKBP52-containing immunophilins are important for the binding of p53 and dynein (Galigniana et al, 2004). In this study, we showed that the activity of p53 is enhanced by the depletion of FKBP52. In this study, the effect of FKBP52 depletion on p53 nuclear

FKBP52 is involved in translocation and transcriptional activity of NFAT

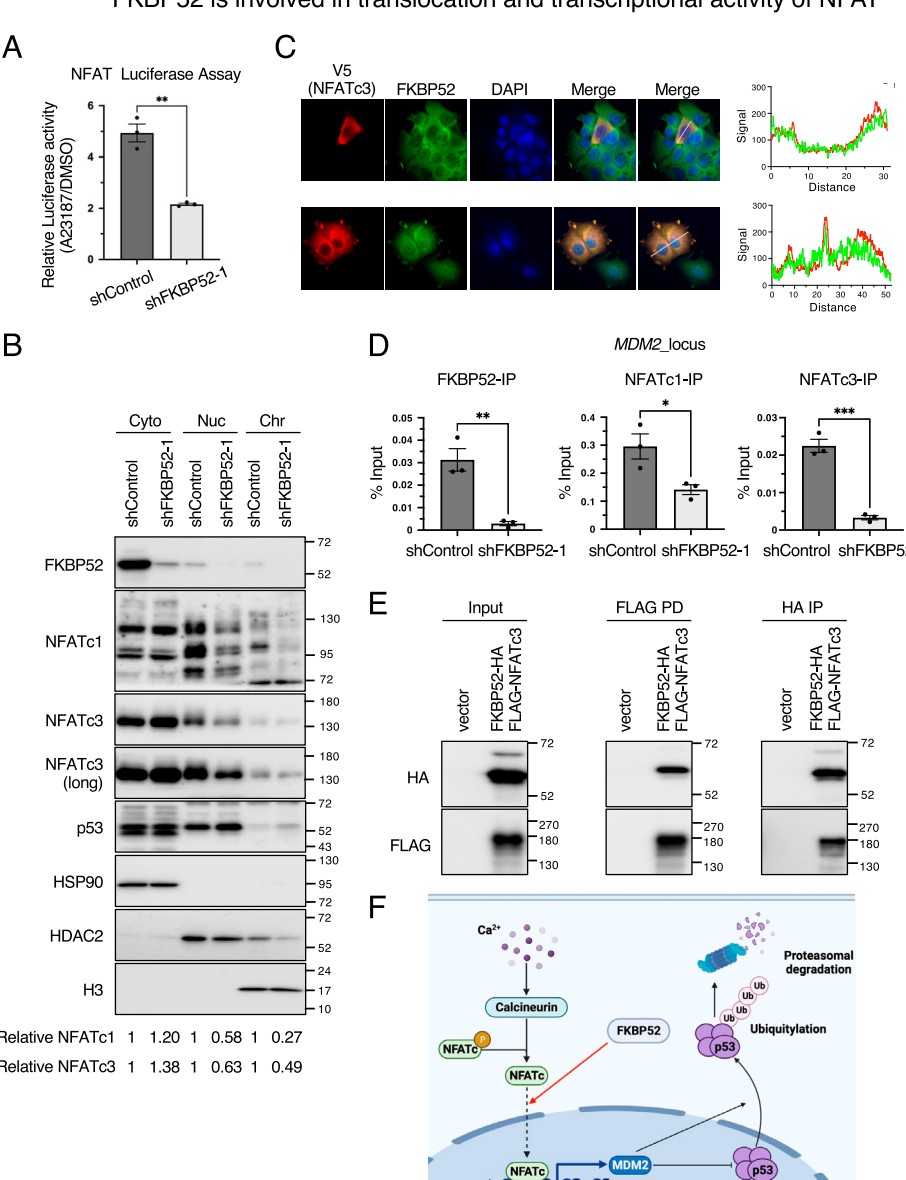

**Figure 5. FKBP52 promotes NFAT transcriptional activity by promoting nuclear translocation.** **(A)** HEK293T cells were transfected with pNL containing the NFAT response sequence and pGL4.51 containing the CMV promoter, in the presence of Dox for 3 d. NanoLuc luciferase activity was normalized to the FLuc activity. A23187 was added at a concentration of 1 μM 8 h before measurement. The bar chart shows the mean ± SEM of three independent experiments. Values were tested using a *t* test. **P < 0.01. **(B)** MCF7 cells were cultured in the presence of Dox for 3 d to knock down FKBP52 or luciferase (shControl) using tetracycline-inducible shRNA. The cells were collected, and the cytoplasm, nucleus, and chromatin extracts were subjected to immunoblotting using the indicated antibodies. Signals were quantified using Image Lab software. HSP90, HDAC2, and H3 were used as loading controls for the cytoplasm, nucleus, and chromatin, respectively. **(C)** Immunofluorescence of V5 and FKBP52 in MCF7 cells overexpressing V5-NFATc3. The thin bars indicate the cross-section of the quantities. The thick bar is the scale bar and indicates 30 μm. The graph on the right shows the signal intensities of V5 and FKBP52 in the cross-section shown in the figure. **(D)** MCF7 cells were cultured in the presence of Dox for 3 d to knock down FKBP52 or luciferase (shControl) using tetracycline-inducible shRNA. ChIP assays were performed as described in the Materials and Methods section. The results are shown as fold enrichment and are expressed as a percentage of the total input chromatin. Data are presented as the mean ± SEM of at least three independent experiments. The bar chart shows the mean ± SEM of three independent experiments. Values were tested using a *t* test. *P < 0.05, **P < 0.01, ***P < 0.001. **(E)** HEK293T cells were transfected with pcDNA3 control vector, or pcDNA3-FKBP52-HA and pcDNA3-FLAG-NFATc3. Immunoprecipitation with FLAG or HA was performed and subjected to immunoblotting using the indicated antibodies. **(F)** Schematic diagram of FKBP52 regulating p53. FKBP52 promotes the nuclear translocation of NFAT, which in turn promotes *MDM2* transcription and p53 degradation.

translocation was not evaluated, but p53 transcriptional activity was increased. Thus, even if FKBP52 depletion suppresses p53 nuclear translocation, the effect of increased p53 expression may be greater than its effect on the inhibition of nuclear translocation. Alternatively, when FKBP52 is depleted, its function in nuclear translocation may be complemented by other immunophilins, such as Cyp-40 and PP5. Our group reported that FKBP52 promotes ERα stabilization in MCF7 cells (Habara et al, 2022). However, the extent of the effect of ERα destabilization caused by FKBP52 depletion on proliferation is unclear. In this study, we confirmed that FKBP52 depletion suppressed cell proliferation and that this phenotype was recovered by p53 depletion. This suggests that at least in MCF7 cells, growth inhibition by FKBP52 depletion may be p53-dependent.

Therefore, the direct effect of ERα destabilization by FKBP52 depletion on cell proliferation may be limited. In addition, because ERα has been reported to inhibit p53 (Liu et al, 2006), FKBP52 depletion may promote p53 activation through two pathways: ERα destabilization and inhibition of the NFAT/MDM2 pathway.

NFATc2, a member of the NFAT family, promotes *MDM2* transcription in a p53-independent manner (Zhang et al, 2012). Depletion of NFATc2 in p53-depleted PC9 cells decreased MDM2 protein expression. However, in p53-depleted HCT116 cells, NFATc2 depletion did not affect MDM2 protein expression without ionomycin, a calcineurin activator. Similarly, in the present study, the depletion of NFATc2 in MCF7 cells did not affect the protein expression of MDM2. This suggests that the amount of NFATc2 protein

endogenously expressed in MCF7 and HCT116 cells may have a limited effect on *MDM2* transcription. We concluded that NFATc1 and NFATc3 are involved in *MDM2* transcription instead of NFATc2 in MCF7 cells. We also demonstrated that calcineurin, which activates NFAT, is involved in *MDM2* transcription. In particular, CaN Aα depletion decreased MDM2 expression more strongly than NFAT depletion. This may be because inhibition of calcineurin affects the overall NFAT family. However, calcineurin dephosphorylates and regulates many targets in a variety of ways, including protein stability and transcriptional activity (Masaki & Shimada, 2022; Masaki et al, 2023a, 2023b). Notably, we recently reported that FOXO1 activates *MDM2* transcription and that calcineurin dephosphorylates and stabilizes FOXO1 (Tomiyasu et al, 2024). These findings suggest that two transcription factors, NFATc and FOXO1, downstream of calcineurin, promote *MDM2* transcription.

In this study, we showed that the depletion of FKBP52 inhibits the nuclear translocation of NFAT. The $Ca^{2+}$/calcineurin pathway is important for the nuclear translocation of NFAT, and the dephosphorylation of NFAT by calcineurin exposes the NLS, which triggers nuclear translocation (Mognol et al, 2016). Although phosphorylation of NFAT is known to cause a band shift (Gwack et al, 2006), depletion of FKBP52 did not cause a band shift of NFAT. This suggests that depletion of FKBP52 does not significantly affect the phosphorylation state of NFAT. FKBP52 associates with NFATc3 and is located in the promoter region of *MDM2*, suggesting that FKBP52 forms a complex with NFAT, translocates to the nucleus, and binds to the promoter region of *MDM2*. Although FKBP52 is known to associate with dynein to promote nuclear translocation of transcription factors, no report has clarified the relationship between NFAT and dynein. Elucidation of this relationship would provide insights into the mechanism by which FKBP52 is involved in the nuclear translocation of NFAT. FKBP52 activates ion channel TRPC1 (Shim et al, 2009) and suppresses TRPC3 (Bandleon et al, 2019) and TRPV5 (Gkika et al, 2006). Given that only TRPC1 is expressed in MCF7 cells (Uhlen et al, 2015) (https://www.proteinatlas.org/ENSG00000144935-TRPC1/cell+line), FKBP52 depletion may inhibit $Ca^{2+}$ influx by suppressing TRPC1 activity. As a result, the $Ca^{2+}$/calcineurin pathway may be suppressed, leading to the inhibition of the nuclear translocation of NFAT.

MCF7 cells have higher $Ca^{2+}$ concentrations than MCF10A cells (Pottle et al, 2013) with calcium channels Orai1 (McAndrew et al, 2011), Orai3 (Faouzi et al, 2011), and TRPM7 (Guilbert et al, 2009) being overexpressed in MCF7 cells. Based on these findings, it is possible that $Ca^{2+}$ signaling is activated in MCF7 cells because of elevated $Ca^{2+}$ concentrations associated with the overexpression of calcium channels. In pancreatic cancer cells, inhibition of the NFAT pathway using cyclosporin A (CsA) results in the suppression of c-myc transcription, leading to an increase in the G1 phase and a decrease in the S phase (Buchholz et al, 2006). In vascular smooth muscle cells, inhibition of the NFAT pathway with CsA or VIVIT also inhibits the transition from the G1 phase to the S phase induced by PDGF-BB (Karpurapu et al, 2008, 2010). In lung cancer cells, NFATc1 transcribes the DNA damage–induced apoptosis suppressor, which in turn inhibits cisplatin-induced apoptosis (Im et al, 2016). In addition, treatment of MCF7 cells with amlodipine, a calcium channel blocker, reduces the abundance of Bcl-2 (Alqudah et al, 2022). These results suggest that the $Ca^{2+}$/NFAT pathway may play a suppressive role in the regulation of apoptosis.

In neuronal cells, FKBP52 inhibits tubulin polymerization (Chambraud et al, 2007, 2010). In addition, the inhibition of tubulin polymerization has been reported to inhibit the nuclear translocation of NFAT (Mackenzie & Oteiza, 2007). Given these facts, it is possible that the depletion of FKBP52 could disrupt the balance between appropriate polymerization and depolymerization of tubulin, thereby inhibiting the nuclear translocation of NFAT. The relationship between tubulin polymerization and FKBP52 in neuronal cells should be investigated in the future.

FKBP52 has three main domains: FK1, FK2, and TPR. The FK1 domain is involved in prolyl isomerase activity, and the TPR domain is involved in binding to HSP90. FKBP52 interacts with its target molecules via one or more of these domains (Hanaki & Shimada, 2023). Because there are no reports of NFAT binding to HSP90, and no binding between NFAT and HSP90 is evident in the interactome database (Oughtred et al, 2021) (https://thebiogrid.org), it is unlikely that FKBP52 binds to NFAT via HSP90. As no molecule has been reported to bind FK2 alone, it may bind FK1 or TPR alone or through multiple domains. As FK506, an inhibitor of the enzymatic activity of FKBP52, does not affect the binding of FKBP52 to dynein (Galigniana et al, 2001), enzymatic activity may not be required for the regulation of nuclear translocation through dynein. However, even though RelA is translocated into the nucleus via dynein, the enzymatic activity of FKBP52 is required to enhance the transcriptional activity of NF-κB (Erlejman et al, 2014). Because this study did not identify the requirement of FKBP52 enzymatic activity, the requirement of HSP90, or the binding domain between NFAT and FKBP52 for the nuclear translocation of NFAT, these are issues to be clarified in the future.

Our results showed that FKBP52 induced p53 degradation and promoted cell proliferation through the NFAT/MDM2 pathway. FKBP52 interacts with a variety of targets and plays a complex role in cell proliferation. This study provides a part of the pathway involved in FKBP52 and helps understand the comprehensive association between FKBP52 and cell proliferation.

# Materials and Methods

### Cell culture and reagents

MCF7 (HTB-22; ATCC) and HEK293T (632180; Takara) cells were cultured in DMEM (044-29765; Wako) supplemented with 10% FBS (173012; Sigma-Aldrich) and antibiotic–antimycotic solution (15240062; Thermo Fisher Scientific). HCT116 (CCL-247; ATCC) cells were cultured in McCoy's 5A medium (16600-082; Gibco) supplemented with 10% FBS and antibiotics. Each cell line was maintained at 37°C in an atmosphere containing 5% $CO_2$.

### Chemicals

The following drugs were used during the course of the study: cycloheximide (037-20991; Wako), A23187 (C7522; Sigma-Aldrich), and INCA-6 (ab145864; Abcam). Cycloheximide was dissolved in water, whereas the others were dissolved in DMSO. The concentrations and durations of treatment are indicated in the legends of the corresponding figures.

## Construction of the expression vectors, and transient transfection

For NFATc3 overexpression in cells, the human NFATc3 corresponding open reading frame (NM_173165.3) was cloned into the N-terminally 3×FLAG-tagged pcDNA3 vector. To create a plasmid with constitutively active NFATc3, the corresponding mutation sites in constitutively active NFATc1 (Neal & Clipstone, 2001) were mutated. The mutation sites are listed as follows: S165A, S168A, S169A, S172A, S174A, S177A, S180A, S181A, S184A, S186A, S207A, S211A, S215A, S236, S244, S288A, S292A, S296A, S300A. For FKBP52 overexpression in cells, the human FKBP52 corresponding open reading frame (NM_002014.4) was cloned into the C-terminally HA-tagged pcDNA3 vector. Transient transfection of the expression vectors was performed using polyethyleneimine (PEI) MAX (24765-1; Polysciences). 293T cells were transfected with a mixture of 3 $\mu$g of a pcDNA3 plasmid and 6 $\mu$g of PEI. The medium was changed 8 h after transfection, and the cells were collected 2 d later.

## Lentivirus generation and infection

Lentivirus generation and infection were performed as previously described (Habara et al, 2022). Briefly, lentiviruses expressing shControl, shFKBP52-1, shFKBP52-2, shFKBP52-3, shFKBP52-4, shNFATc1-1, shNFATc1-2, shNFATc2-1, shNFATc2-2, shNFATc3-1, shNFATc3-2, shCaN A$\alpha$-1, shCaN A$\alpha$-2, and shp53 were generated by the co-transfection of HEK293T cells with 1.54 $\mu$g psPAX2 and 0.86 $\mu$g pMD2.G, and the 2 $\mu$g CS-RfA-ETBsd or CS-RfA-ETHyg, using 8.8 $\mu$g PEI. 2 d after transfection, the virus was collected and used to infect MCF7 or HCT116 cells. The cells were incubated with the lentivirus for 24 h. Virus-infected cells were treated with 10 $\mu$g/ml blasticidin (A1113903; Gibco) or 200 $\mu$g/ml hygromycin (084-07681; Wako) for 2 d. To drive shRNA expression, doxycycline (Dox; D9891; Sigma-Aldrich) was added to the medium at a concentration of 1 $\mu$g/ml. The target sequences of the shRNAs are listed in Table S2. For NFATc3 overexpression in cells, the corresponding human NFATc3 open reading frame (NM_173165.3) was cloned into the NH2-terminally V5-tagged CSII-CMV-MCS-IRES2-Bsd vector. If not stated, shFKBP52-1 was used to knock down FKBP52.

## Immunoblotting

Immunoblotting was performed as previously described (Hanaki et al, 2021). To prepare total cell lysates, the collected cells were washed with ice-cold PBS, suspended in sample buffer (2% SDS, 10% glycerol, 100 $\mu$M dithiothreitol, 0.1% bromophenol blue, and 50 mM Tris–HCl at pH 6.8), and boiled for 5 min. Raw digital images were captured using ChemiDoc Imaging System (Bio-Rad). The bands of the target protein were quantified using Image Lab (Bio-Rad) and normalized to those of $\beta$-actin or HSP90. Representative images are shown in the figures. The decay curve of p53 was plotted using GraphPad Prism version 9 (GraphPad Software), based on the band intensity of HSP90. The exponential one-phase decay equation from nonlinear regression was used to generate the decay curve. All the antibodies used in this study are listed in Table S3.

## Immunoprecipitation

Immunoprecipitation was performed as previously described (Masaki et al, 2021). Cells were lysed in immunoprecipitation kinase buffer (50 mM Hepes–NaOH at pH 8.0, 150 mM NaCl, 2.5 mM ethylene glycol-bis [$\beta$-aminoethyl ether]-N,N,N′,N′-tetraacetic acid, 1 mM dithiothreitol, 0.1% Tween-20, and 10% glycerol) supplemented with protease inhibitors (E-64, leupeptin, pepstatin A, and aprotinin) and phosphatase inhibitors (50 mM NaF, 0.1 mM Na3VO4, 15 mM p-nitrophenylphosphate, and 80 mM $\beta$-glycerophosphate). The lysates were incubated with FLAG M2 agarose (A2220; Sigma-Aldrich), or immunoprecipitation was performed with an HA-tag antibody for 1 h at 4°C with rotation.

## RT–qPCR

Total RNA was extracted as previously described (Shimada et al, 2008). Briefly, total RNA was extracted using ISOGEN II (311-07361; Nippon Gene), according to the manufacturer's protocol, and reverse transcription was performed. RNA was reverse-transcribed with random primers using High-Capacity cDNA Reverse Transcription Kit (4368814; ABI). qPCR was performed using FastStart Universal SYBR Green Master Mix (11226200; Roche) and StepOnePlus Real-Time PCR System (Applied Biosystems). The expression levels were normalized to those of glyceraldehyde 3-phosphate dehydrogenase (GAPDH), TATA-box–binding protein, and ACTB. The target sequence of each gene is listed in Table S4.

## Luciferase reporter assay

A luciferase reporter assay was performed as previously described (Maeda et al, 2022). pNL(NlucP/p53-RE/Hygro) (CS194102; Promega) was used for measuring the transcriptional activity of p53. To investigate the promoter activity of MDM2, the promoter region of MDM2 (–133 to +33) indicated in the previous study (Zhang et al, 2012) was subcloned upstream of NLuc. The pNL plasmid with the NFAT response sequence and NanoLuc downstream of the response sequence, the pGL4.51 plasmid with the CMV promoter and firefly luciferase (FLuc) downstream of the promoter, and the pGL4.53 plasmid with the PGK promoter and FLuc downstream of the promoter were kindly provided by Dr. Ichiro Yamamoto. 293T cells were cultured at a density of $2 \times 10^4$ cells/well in 96-well plates for 24 h in MEM Alpha (41061-029; Gibco) supplemented with 5% FBS and 1 $\mu$g/ml Dox. The cells were transfected with pNL containing the p53 response sequence and pGL4.53, or pNL containing the NFAT response sequence and pGL4.51, or pNL containing the MDM2 promoter sequence and pGL4.53 using PEI. When measuring the promoter activity of MDM2, constitutively active NFATc3 was also simultaneously transfected. 2 d after transfection, cells were treated with 1 $\mu$M A23187 or DMSO for 8 h. Luminescence caused by FLuc and NanoLuc luciferase was measured using the Nano-Glo Dual-Luciferase Reporter Assay System (N1610; Promega) and Nivo (PerkinElmer).

## Cell cycle analysis

Collected cells were washed with PBS and fixed with 70% ethanol. After fixation, the cells were washed again with PBS and incubated with PBS containing 50 μg/ml propidium iodide (PI) and 100 μl/ml RNase for 1 h at 37°C. Flow cytometry was performed on a FACSVerse flow cytometer (BD Biosciences). The cell cycle profile was analyzed using BD FACSuite software v1.0.6 (BD Biosciences).

## ChIP-qPCR analysis

ChIP-qPCR analysis was performed as previously described (Masaki et al, 2021). Cells were fixed with 2 mM ethylene glycol-bis-succinimidyl succinate (21565; Thermo Fisher Scientific) for 30 min and then with 0.5% formaldehyde for 5 min at room temperature. Fixation was terminated by the addition of glycine at a final concentration of 125 mM and incubation for an additional 5 min. The cells were isolated by centrifugation and lysed for 10 min at 4°C in 800 μl of buffer LB1 (5 mM Hepes–NaOH at pH 8.0, 200 mM KCl, 1 mM CaCl2, 1.5 mM MgCl2, 5% sucrose, and 0.5% Nonidet P-40) supplemented with protease inhibitors (phenylmethylsulfonyl fluoride, leupeptin, pepstatin A, and aprotinin) and subjected to 50 30-s bursts of ultrasonic treatment with a Bioruptor BR-II instrument (Sonic Bio) to obtain DNA fragments of 300–500 bp. Antibodies against FKBP52, NFATc1, or NFATc3 (Table S3) were bound to Dynabeads (10004D; Invitrogen), incubated overnight at 4°C in LB1 supplemented with 10% BSA, and then added to the samples. After overnight incubation at 4°C, the beads were isolated and washed consecutively with Wash Buffer 1 (5 mM Hepes–NaOH at pH 8.0, 200 mM KCl, 1 mM CaCl2, 1.5 mM MgCl$_2$, 5% sucrose, and 0.5% Nonidet P-40), Wash Buffer 2 (5 mM Hepes–NaOH at pH 8.0, 500 mM KCl, 1 mM CaCl$_2$, 5% sucrose, and 0.5% Nonidet P-40), and Wash Buffer 3 (10 mM Tris–HCl at pH 8.0, 1 mM EDTA). DNA was eluted from the beads in elution buffer (50 mM Tris–HCl at pH 8.0, 10 mM EDTA, and 1% SDS), and cross-links were reversed by incubation overnight at 65°C. DNA was eluted from the beads in elution buffer (50 mM Tris–HCl at pH 8.0, 10 mM EDTA, and 1% SDS), and cross-links were reversed by incubation overnight at 65°C. DNA was purified using LaboPass PCR Purification Kit (CMR0112; CosmoGenetech) and subjected to qPCR analysis using the primers listed in Table S4.

## Subcellular fractionation

The collected cells were fractionated into their respective fractions using Subcellular Protein Fractionation Kit for Cultured Cells (78840; Thermo Fisher Scientific). All steps were performed according to the manufacturer's protocol. Briefly, the collected cells were washed with PBS and suspended in CEB containing protease inhibitors. The cells were gently mixed at 4°C for 10 min, and centrifuged at 500g for 5 min, and the supernatant was collected as the cytosolic fraction. MEB containing protease inhibitors was added to the cell pellet, vortexed for 5 s, and gently mixed at 4°C for 10 min. After centrifuging at 3,000g for 5 min, the supernatant was collected as the membrane fraction. NEB containing protease inhibitors was added to the cell pellet, vortexed for 15 s, and gently mixed at 4°C for 30 min. After centrifuging at 5,000g for 5 min, the supernatant

was collected as the soluble nuclear fraction. The pellet remaining after centrifugation was used as the chromatin fraction.

## Immunofluorescence

MCF7 cells overexpressing V5-NFATc3 were cultured on coverslips for 2 d. After washing the cells with PBS, they were fixed with 4% PFA in PBS for 10 min. After fixation, the cells were washed with PBS and then permeabilized with 0.1% Triton X-100/PBS for 5 min. After washing with PBS, cells were incubated in 0.2% gelatin/PBS for 1 h, washed with PBS, and incubated with rabbit anti-FKBP52 and mouse anti-V5 antibodies (1:100) for 1 h. Then, goat anti-rabbit IgG (H+L)–Alexa Fluor 488 (Thermo Fisher Scientific), and goat anti-mouse IgG (H+L)–Alexa Fluor 568 (Thermo Fisher Scientific) were diluted (1:200) and incubated with the cells for 30 min. Coverslips were mounted using ProLong Diamond Antifade Mountant with DAPI (Thermo Fisher Scientific). Confocal images were taken at a magnification of 63× using a Mica Microhub system (Leica Microsystems). Signal quantification was conducted using ImageJ (Abramoff et al, 2004; Schneider et al, 2012) (https://imagej.net/ij/), and graphs were created using GraphPad Prism 6 (GraphPad Software Inc.). Rabbit anti-FKBP52 and mouse anti-V5 antibodies are listed in Table S3.

## Measuring calcium concentration

A recording buffer containing 138 mM NaCl, 5 mM KCl, 1 mM MgCl$_2$, 5 mM glucose, 10 mM Hepes (pH 7.6), 1.5 mM CaCl$_2$, and 0.1% BSA was prepared. MCF7 cells were seeded at a density of $1.5 \times 10^4$ cells per well in a 96-well plate. 2 d later, to prepare Fura-2AM loading buffer, Fura-2AM (F015; Dojindo) was dissolved in the recording buffer at a concentration of 5 μM and sonicated. The cells were incubated with Fura-2AM loading buffer for 1 h at room temperature. After incubation, the cells were washed twice with a recording buffer. Before measurement, the recording buffer was replaced with a fresh recording buffer, and the measurement was started 5 min after the replacement. The excitation wavelengths were set at 340 and 380 nm, whereas the emission was measured at 510 nm. The measurement was performed using an ARVO X4 plate reader (Revvity).

## Candidate isolation

The list of transcription factors that bind to the promoter region of *MDM2* was obtained from the GeneCards website; the GeneHancer ID is GH12J068803 (Fishilevich et al, 2017) (https://www.genecards.org). Transcription factors whose knockdown decreases *MDM2* mRNA expression were obtained using KnockTF (Feng et al, 2020) (https://www.licpathway.net/KnockTFv1/). Correlation coefficients of *MDM2* expression were calculated using TCGA PANCAN database (Cancer Genome Atlas Research Network et al, 2013).

## Bioinformatics analysis

Publicly available RNA-seq data for FKBP52 KD cells were obtained from the DNA Data Bank of Japan Sequence Read Archive under

accession number DRA011728 (Habara et al, 2022). Gene set enrichment analysis (GSEA) was performed with Signal2Noise values for all detected genes for the indicated comparisons as the ranking metric using GSEA software, version 4.2.0 (Mootha et al, 2003; Subramanian et al, 2005). A volcano plot was generated using DEBrowser v1.24.1 (Kucukural et al, 2019).

## Correlation coefficient analysis

The relative abundance values were obtained from the following website: https://proteomics.broadapps.org/CPTAC-BRCA2020/ (Krug et al, 2020). Correlation coefficients were calculated, and scatter plots were created using GraphPad Prism 6 (GraphPad Software Inc.).

## Statistical analysis

A two-sided $t$ test was used to compare the two groups (Figs 1C–E, 3B and C, 4B and E, 5A and D, S1C, S2F, S3B, S4B, and S5C). To compare three or more groups, one-way ANOVA followed by Dunnett's multiple comparisons test was used (Figs 2B and C and 4C, F, and G and S2G). The results were considered statistically significant at *$P$ < 0.05, **$P$ < 0.01, ***$P$ < 0.001, and ****$P$ < 0.0001. Statistical analyses were performed using GraphPad Prism 6 (GraphPad Software Inc.).

# Supplementary Information

# Acknowledgements

We thank Ms. Naoko Kawasaki for providing technical assistance and Dr. Ichiro Yamamoto for materials. We thank Dr. Tomohiro Yamashita and Dr. Shigeaki Sugiyama for providing technical advice. We also thank the Yamaguchi University Project for the formation of the Core Research Center. This work was supported by the Japan Society for the Promotion of Science (JSPS) KAKENHI (grant numbers 20K21503 and 21H02403 to M Shimada), Canon Medical Systems Corporation (S22-0047 to M Shimada), and Fusion Oriented REsearch for disruptive Science and Technology (JPMJFR2065 to M Shimada).

## Author Contributions

S Hanaki: conceptualization, formal analysis, investigation, visualization, and writing—original draft, review, and editing.
M Habara: formal analysis, investigation, and writing—review and editing.
H Tomiyasu: investigation.
Y Sato: investigation.
Y Miki: investigation.
T Masaki: investigation.
S Shibutani: investigation.
M Shimada: conceptualization, formal analysis, funding acquisition, investigation, visualization, project administration, and writing—original draft, review, and editing.

## Conflict of Interest Statement

The authors declare that they have no conflict of interest.

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
