## [Reviewer comments · Life Science Alliance]

Life Science Alliance

NFAT Activation by FKBP52 Promotes Cancer Cell Proliferation by Suppressing p53

Shunsuke Hanaki, Makoto Habara, Haruki Tomiyasu, Yuki Sato, Yosei Miki, Takahiro Masaki, Shusaku Shibutani, and Midori Shimada

DOI: <https://doi.org/10.26508/lsa.202302426>

Corresponding author(s): Midori Shimada, Nagoya University

Review Timeline:

Submission Date:	2023-10-09
Editorial Decision:	2023-11-27
Revision Received:	2024-05-01
Editorial Decision:	2024-05-03
Revision Received:	2024-05-07
Accepted:	2024-05-07

Transaction Report:

November 27, 2023

Re: Life Science Alliance manuscript #LSA-2023-02426-T

Prof. Midori Shimada
Yamaguchi University
Joint Faculty of Veterinary Science
1677-1 Yoshida
Yamaguchi, Yamaguchi 753-8511
Japan

Dear Dr. Shimada,

Thank you for submitting your manuscript entitled "NFAT Activation by FKBP52 Promotes Cancer Cell Proliferation by Suppressing p53" to Life Science Alliance. The manuscript was assessed by expert reviewers, whose comments are appended to this letter. We invite you to submit a revised manuscript addressing the Reviewer comments.

Thank you for this interesting contribution to Life Science Alliance. We are looking forward to receiving your revised manuscript.

Sincerely,

B. MANUSCRIPT ORGANIZATION AND FORMATTING:

Reviewer #1 (Comments to the Authors (Required)):

This study provides of role for FKBP52 in the regulation of TP53 expression in TP53 wild type cancer cells. The data is clearly presented and the manuscript is for the most part well written. However, additional experiments are needed to define the mechanism of regulation of TP53-MDM2 axis by FKBP52 and translational significance of the study. First, key shRNA knockdown including Figures 1C, 2A, 2B, 3A, 4B, 4C, 4D, 5B and 5C should be repeated using a second targeting sequence. Second, the authors should determine the localization of TP53 and protein expression of other target genes in MCF7 cells. Third, the protein expression of TP53 and p21 in HCT116 cells should be added to the manuscript. Fourth, key experiments should be repeated in these cells (e.g., data from Figures 2, and 5). And fifth, all western blots should include two molecular markers flanking the relevant bands. Also, the quality of some of the blots should be improved including HSP90 from Fig 1C, 5B and EV1; TP53 from EV; and betaActin in Figure 4B and 4D. Finally, the specificity of the FKBP52 antibody should be defined using two shRNA and perhaps a second antibody from a different company. This is particularly important as western blots for this protein show different signal throughout the manuscript

Reviewer #2 (Comments to the Authors (Required)):

In this manuscript, the authors uncovered the link between NFAT and p53 pathways by revealing the novel role of FKBP52 in regulation of NFAT translocation. The authors also showed that NFAT regulates p53 stability by inducing transcription of Mdm2, a ubiquitin ligase that targets p53. The novelty of this study resides in the role of FKBP52 in regulation of NFAT and p53. FKBP52 is known to be involved in cell proliferation; however, the detailed mechanism remains poorly understood. However, this study's novelty is decreased by the previous findings showing the role of NFAT in regulation of Mdm2 promoter (Xu Zhang et al., JBC, 2012). Also, the current study does not present any in vivo study; hence, the physiological meaning of this finding has not been determined. The conclusion seems reasonable, but more depth needs to be presented in the mechanistic aspect.

1. In Fig. 1A and B, I recommend showing the full analysis of RNA-seq to understand the comprehensive role of FKBP52 and emphasize the significance of the p53 pathway.
2. In Fig. 2A, in addition to the cell culture system, tumor cell growth needs to be shown in a physiological setting.
3. In Fig. 4, Mdm2 promoter reporter assay was used to demonstrate the role of NFAT. To clearly demonstrate the direct role of NFAT, it is better to use constitutively active NFAT instead of ionophore. Also, direct Ca²⁺ measurement is needed to show that these effects are not derived from changes in intracellular Ca²⁺ levels, especially considering the role of FKBP52 in regulation of Trpc3 channels.
4. In Fig. 5, the nuclear fraction does not show any FKBP52 in the nucleus in biochemical assays, but ChIP results showed that it is associated with NFAT in the Mdm2 promoter. These conflicting results need to be resolved. I also recommend using confocal analysis to show the co-localization of NFAT and FKBP52.
5. In the Discussion section, the potential upstream Ca²⁺ signaling that regulates NFAT activation in their experimental setting needs to be discussed. Also, references indicating the role of the Ca²⁺-NFAT pathway in cell death and cell cycle arrest need to be included.
6. It is a minor point, but the molecular weight should be presented in the immunoblotting analysis throughout the figures.

Reviewer 1

[Response] We thank the reviewer for the careful review of our manuscript. We also thank the reviewer for the constructive suggestions which have helped us to considerably improve our manuscript. Our specific responses to the points raised can be found below (this letter contains low-resolution thumbnails for clarity; please refer to the manuscript for high-resolution figures).

1) First, key shRNA knockdown including Figures 1C, 2A, 2B, 3A, 4B, 4C, 4D, 5B and 5C should be repeated using a second targeting sequence.

[Response] As suggested by the reviewer, we performed shRNA knockdown experiments using the second or third target sequence.

Figure 1C: We have shown the effect of shFKBP52-2 on p53 and p21 expression in the original manuscript (Figure 1C). Similar to the effect of shFKBP52-1, depletion of FKBP52 by shFKBP52-2 increased the abundance of p53 and p21 in MCF7 cells. We have now addressed this point in the revised manuscript (page 5, lines 89-91).

Figure 2A: Given that it was technically difficult, for unknown reasons, to deplete FKBP52 by shFKBP52-2 in addition to shRNA-mediated depletion of p53 in MCF7 cells, we decided to deplete FKBP52 by one of three independent shRNAs (shFKBP52-1, shFKBP52-3, and shFKBP52-4) and CRISPR-Cas9-mediated deletion of the p53 gene in HCT116 cells (clone 10) (new Supplementary Figure S2A and new Supplementary Figure S2B). These experiments revealed that the attenuation of cell proliferation caused by FKBP52 deletion was restored by the additional deletion of p53 in HCT116 cells (new Supplementary Figure S2C). We have now addressed these points in the revised manuscript (page 6, lines 110-115).

Figure 2B: This experiment was repeated using shFKBP52-3. Similar to the results with shFKBP52-1, depletion of FKBP52 resulted in an increase in the sub-G1 population, which was restored by the additional depletion of p53 (new Supplementary Figure S2E). We have now addressed this point in the revised manuscript (page 6, lines 117-119).

Figure 3A: This experiment was repeated using shFKBP52-3. Similar to the results obtained with shFKBP52-1, p53 was stabilized by FKBP52 depletion (new Supplementary Figure S3A). We have now addressed this point in the revised manuscript (page 7, lines 135-137).

Figure 4B: The original results with shNFATc1, shNFATc3, and shNFATc2 showed that the abundance of MDM2 was reduced by the depletion of NFATc1 and NFATc3, but not NFATc2. Thus, we repeated these experiments with other sets of shRNAs (shNFATc1-2, shNFATc3-2, and shNFATc2-2) and obtained similar results (new Supplementary Figure S4A). We have now addressed this point in the revised manuscript (page 8, lines 164-166).

Figure 4C: We also used shNFATc1-2 and shNFATc3-2, and obtained similar results (new Supplementary Figure S4B). We have now addressed this point in the revised manuscript (page 8, lines 164-165).

Figure 4D: CaN depletion by shCaN A α -1 reduced the abundance of MDM2 in the original manuscript. We repeated this experiment with another shRNA (shCaN A α -2), and obtained similar results (new Supplementary Figure S4C). We have now addressed this point in the revised manuscript (page 8, lines 168-170).

Figure 5B: The original experiments revealed that the nuclear translocation of NFATc1 and NFATc3 was inhibited by FKBP52 depletion mediated by shFKBP52-1. As suggested by the reviewer, we have performed the same experiments using shFKBP52-3. The effect of shFKBP52-3 was similar to that of shFKBP52-1 (new Supplementary Figure S5A). We have now addressed this point in the revised manuscript (page 9, lines 184-188).

2) Second, the authors should determine the localization of TP53 and protein expression of other target genes in MCF7 cells.

[Response] The subcellular localization of p53 was examined by the biochemical fractionation of FKBP52-depleted cells. These results revealed that the abundance of p53 was increased by FKBP52 depletion; however, its localization was not affected (new Figure 5B). We have now addressed this point in the revised manuscript (page 9, lines 190-191).

[Response] The protein expression of GADD45A and PUMA, which are other targets of p53, was examined in MCF7 cells depleted of FKBP52. We found that the abundance of GADD45A and PUMA was increased in FKBP52-depleted cells mediated by two independent shRNA sequences (shFKBP52-1 and shFKBP52-2) (new Supplementary Figure S1A). We have now addressed this point in the revised manuscript (page 5, lines 91-92).

3) Third, the protein expression of TP53 and p21 in HCT116 cells should be added to the manuscript.

[Response] We have already shown the data requested by the reviewer in the original manuscript (Supplementary Figure S1B). In HCT116 cells, FKBP52 depletion increased the abundance of p53 and p21. We have now addressed this point in the revised manuscript (page 5, lines 94-95).

4) Fourth, key experiments should be repeated in these cells (e.g., data from Figures 2, and 5).

[Response] As suggested by the reviewer, we conducted key experiments using the HCT116 cells.

Figure 2A: FKBP52 depletion inhibited the proliferation of HCT116 cells, similar to that of MCF7 cells. In addition, combined depletion of FKBP52 and p53 recovered the proliferation of HCT116 cells compared with FKBP52-depletion alone (new Supplementary Figure S2D). We have now addressed this point in the revised

manuscript (page 6, lines 115-116).

As mentioned above, we also conducted experiments similar to the original Figure 2A in HCT116 cells. In this case, we depleted FKBP52 using one of three independent shRNAs (shFKBP52-1, shFKBP52-3, and shFKBP52-4) and CRISPR-Cas9-mediated deletion of the p53 gene (new Supplementary Figures S2A and S2B). These experiments revealed that the attenuation of cell proliferation caused by deletion of FKBP52 was restored by the additional deletion of p53 in HCT116 cells (new Supplementary Figure S2C). We have now addressed this point in the revised manuscript (page 6, lines 114-115).

Figure 2C: As shown in MCF7 cells, we confirmed that the abundance of *CDKN1A* was increased by FKBP52 depletion in HCT116 cells, (new Supplementary Figure S2G). We have now addressed this point in the revised manuscript (page 6, lines 126-127).

Figure 5B: Nuclear translocation of NFATc1 and NFATc3 is attenuated by FKBP52 depletion in MCF7 cells. We have now shown the similar results in HCT116 cells (Supplementary Figure S5B). We have now addressed this point in the revised manuscript (page 9, lines 188-189). For Figures 5A and 5D, 293T cells were used because of the importance of transfection efficiency.

5) And fifth, all western blots should include two molecular markers flanking the relevant bands.

[Response] As suggested by the reviewer, molecular weight markers have been added to all western blot figures flanking the relevant bands.

6) Also, the quality of some of the blots should be improved including HSP90 from Fig 1C, 5B and EV1; TP53 from EV; and betaActin in Figure 4B and 4D.

[Response] In response to the reviewer's suggestion, the quality of the blots, including Hsp90 and β -actin, was improved (new Figure 1C, 4B, 4D, 5B, and Supplementary Figure S1B).

7) Finally, the specificity of the FKBP52 antibody should be defined using two shRNA and perhaps a second antibody from a different company. This is particularly important as western blots for this protein show different signal throughout the manuscript.

[Response] We examined two antibodies against FKBP52 (H00002288-M01 from Abnova and 66040-2-Ig from Proteintech). These results indicate that the signal corresponding to FKBP52 detected with either antibody was almost completely reduced in cells lacking FKBP52 (for review purposes only).

Reviewer 2

1) In Fig. 1A and B, I recommend showing the full analysis of RNA-seq to understand the comprehensive role of FKBP52 and emphasize the significance of the p53 pathway.

[Response] As suggested by the reviewer, the full analysis of RNA-seq results is provided in new Supplementary Table 1. We have now addressed this point in the revised manuscript (page 5, lines 85-87).

2) In Fig. 2A, in addition to the cell culture system, tumor cell growth needs to be shown in a physiological setting.

[Response] We agree with the reviewer that xenograft experiments are crucial. In the process of sorting priorities for revision, the editor suggested that data for xenograft experiments would not be needed for further consideration of our manuscript. We apologize for not being able to fulfill this constructive request of the reviewer at this time. Notably, we have already published the tumor-suppressive effect of FKBP52 depletion in vivo using a mouse xenograft (Habara *et al.*, 2022, PNAS).

3-1) In Fig. 4, Mdm2 promoter reporter assay was used to demonstrate the role of NFAT. To clearly demonstrate the direct role of NFAT, it is better to use constitutively active NFAT instead of ionophore.

[Response] As suggested by the reviewer, we performed a luciferase assay using control and FKBP52-depleted cells overexpressing constitutively active NFATc3, which does not undergo phosphorylation and promotes its nuclear translocation. Unexpectedly, depletion of FKBP52 did not affect the promoter activity of MDM2 (for review purposes only). We speculate that FKBP52 promotes NFAT localization, such that overexpressed constitutively active NFATc3 is no longer regulated by FKBP52. In fact, overexpression of NFATc3 in its constitutively activated form showed that the amount of NFATc3 bound to the nucleus and chromatin in FKBP52 knockdown cells was comparable to that in control cells (for review purposes only).

3-2) Also, direct Ca²⁺ measurement is needed to show that these effects are not derived from changes in intracellular Ca²⁺ levels, especially considering the role of FKBP52 in regulation of Trpc3 channels.

[Response] In response to this comment, we measured intracellular Ca²⁺ concentration using Fura 2-AM. First, to determine whether Fura 2-AM indicates a of Ca²⁺ concentration, ATP was treated with Fura 2-AM-incorporated MCF7 cells and fluorescence was measured. In response to ATP, the ratio of 340/380 fluorescence increased, indicating that Fura 2-AM is an indicator of Ca²⁺ concentration in MCF7 cells. Therefore, the intracellular Ca²⁺ concentration was measured, and found that FKBP52 depletion did not affect the intracellular Ca²⁺ concentration (new Supplementary Figure S5C). We have now addressed this point in the revised manuscript (page 9, lines 192-195).

Figure legend (Figure 4 for reviewer)

MCF7 cells were cultured in the presence of Dox for 2 days to knockdown luciferase using tetracycline-inducible shRNA. Fura 2-AM was incorporated and the signal ratio of 340/380 was measured (n=3). After 5 measurements times, ATP (10 M) was added and measured 20 times. The ATP response curve was created

using GraphPad Prism 6 (GraphPad Software Inc., San Diego, CA, USA).

4-1) In Fig. 5, the nuclear fraction does not show any FKBP52 in the nucleus in biochemical assays, but ChIP results showed that it is associated with NFAT in the Mdm2 promoter.

[Response] The signal corresponding to FKBP52 in the nucleus was invisible in the original data, probably due to the low quality of the data. We have now replaced the improved immunoblot analysis with antibodies against FKBP52 (new Figure 5B). These results suggest that FKBP52 is indeed present in the nuclear and chromatin fractions, but only in small amounts (new Figure 5B). We have now addressed these points in the revised manuscript (page 9, lines 189-190).

4-2) I also recommend using confocal analysis to show the co-localization of NFAT and FKBP52.

[Response] As suggested by the reviewer, we performed confocal analysis and found that a subset of FKBP52 was colocalized with NFATc3 (new Figure 5C). We have now addressed these points in the revised manuscript (page 9, lines 191-192).

5) In the Discussion section, the potential upstream Ca²⁺ signaling that regulates NFAT activation in their experimental setting needs to be discussed. Also, references indicating the role of the Ca²⁺-NFAT pathway in cell death and cell cycle arrest need to be included.

[Response] We have added the following discussion on the potential activation of calcium signaling due to the increased Ca²⁺ concentration in the MCF7 cells used in our experiments. We have also described the potential role of the Ca²⁺-NFAT pathway in apoptosis and cell cycle arrest (page 13, lines 281-294): “MCF7 cells have higher Ca²⁺ concentrations than MCF10A cells (Pottle *et al.*, 2013, J. Cancer Ther), with calcium channels Orai1 (McAndrew *et al.*, 2011, Mol Cancer Ther.), Orai3 (Faouzi *et al.*, 2011, J Cell Physiol), and TRPM7 (Guilbert *et al.*, 2009, Am J Physiol Cell Physiol) being overexpressed in MCF7 cells. Based on these findings, it is possible that Ca²⁺ signaling is activated in MCF7 cells because of elevated Ca²⁺ concentrations associated with the overexpression of calcium channels. In pancreatic cancer cells, inhibition of the NFAT pathway using cyclosporin A (CsA) results in the suppression of c-myc transcription, leading to an increase in the G1 phase and a decrease in the S phase (Buchholz *et al.*, 2006, EMBO J). In vascular smooth muscle cells, inhibition of the NFAT pathway with CsA or VIVIT also inhibits the transition from G1 phase to S phase induced by platelet-derived growth factor-BB (PDGF-BB) (Karpurapu *et al.*, 2010, J Biol Chem and Karpurapu *et al.*, 2008, J Biol Chem). In lung cancer cells, NFATc1 transcribes the DNA damage-induced apoptosis suppressor (DDIAS), which in turn inhibits cisplatin-induced apoptosis (Im *et al.*, 2016, Biochim Biophys Acta). In addition, treatment of MCF7 cells with amlodipine, a calcium channel blocker, reduces the abundance of Bcl-2 (Alqudah *et al.*, 2022, Biomed Rep). These results suggest that the Ca²⁺-NFAT pathway may play a suppressive role in the regulation of apoptosis”.

6) It is a minor point, but the molecular weight should be presented in the immunoblotting analysis throughout the figures.

[Response] We have added molecular weight markers to the immunoblot analysis.

May 3, 2024

RE: Life Science Alliance Manuscript #LSA-2023-02426-TR

Prof. Midori Shimada
Nagoya University
Joint Faculty of Veterinary Science
1677-1 Yoshida
Yamaguchi, Yamaguchi 753-8511
Japan

Dear Dr. Shimada,

Thank you for submitting your revised manuscript entitled "NFAT Activation by FKBP52 Promotes Cancer Cell Proliferation by Suppressing p53". We would be happy to publish your paper in Life Science Alliance pending final revisions necessary to meet our formatting guidelines.

- please be sure that the authorship listing and order is correct
- please add ORCID ID for corresponding author--you should have received instructions on how to do so
- please add Keywords for your manuscript in our system
- please add a Category for your manuscript in our system
- please upload all figure files as individual ones, including the supplementary figure files
- please use the [10 author names, et al.] format in your references (i.e. limit the author names to the first 10)

FIGURE CHECKS

-we encourage you to arrange Figure S2 so that the panels are introduced in alphabetical order, and update the legend and callouts accordingly

A. FINAL FILES:

B. MANUSCRIPT ORGANIZATION AND FORMATTING:

Sincerely,

May 7, 2024

RE: Life Science Alliance Manuscript #LSA-2023-02426-TRR

Prof. Midori Shimada
Nagoya University
Department of Molecular Biology, Graduate School of Medicine
65 Tsurumai-cho, Showa-ku
Nagoya, Aichi 466-8550
Japan

Dear Dr. Shimada,

Thank you for submitting your Research Article entitled "NFAT Activation by FKBP52 Promotes Cancer Cell Proliferation by Suppressing p53". It is a pleasure to let you know that your manuscript is now accepted for publication in Life Science Alliance. Congratulations on this interesting work.

DISTRIBUTION OF MATERIALS:

Again, congratulations on a very nice paper. I hope you found the review process to be constructive and are pleased with how the manuscript was handled editorially. We look forward to future exciting submissions from your lab.

Sincerely,
